# Do Circulating Histones Represent the Missing Link among COVID-19 Infection and Multiorgan Injuries, Microvascular Coagulopathy and Systemic Hyperinflammation?

**DOI:** 10.3390/jcm11071800

**Published:** 2022-03-24

**Authors:** Daniela Ligi, Rosanna Maniscalco, Mario Plebani, Giuseppe Lippi, Ferdinando Mannello

**Affiliations:** 1Unit of Clinical Biochemistry, Department of Biomolecular Sciences—DISB, University of Urbino Carlo Bo, 61029 Urbino, Italy; daniela.ligi@uniurb.it (D.L.); r.maniscalco@campus.uniurb.it (R.M.); 2Department of Medicine—DIMED, University of Padua, 35128 Padua, Italy; 3Section of Clinical Biochemistry, University Hospital of Verona, 37134 Verona, Italy

**Keywords:** histone, COVID-19, coagulopathy, cytokine storm, inflammation, multiorgan injury, neutrophil extracellular trap, heparin, heparinoids, laboratory medicine

## Abstract

Several studies shed light on the interplay among inflammation, thrombosis, multi-organ failures and severe acute respiratory syndrome coronavirus 2 (SARS-CoV-2) infection. Increasing levels of both free and/or circulating histones have been associated to coronavirus disease 2019 (COVID-19), enhancing the risk of heart attack and stroke with coagulopathy and systemic hyperinflammation. In this view, by considering both the biological and clinical rationale, circulating histones may be relevant as diagnostic biomarkers for stratifying COVID-19 patients at higher risk for viral sepsis, and as predictive laboratory medicine tool for targeted therapies.

Several studies shed light on the crucial interplay among inflammation, thrombosis, cardiovascular diseases, multi-visceral manifestations, and severe acute respiratory syndrome coronavirus 2 (SARS-CoV-2) infection, namely linking the roles of neutrophil extracellular traps (NETs), nucleosomes, histones, cytokines, and the coagulation cascade [1,2,3,4,5]. A possible novel association of coronavirus disease 2019 (COVID-19) with cardiovascular risk, heart attack, and stroke has also been underpinned [6,7], thus suggesting an urgent need to explore the biomolecular characteristics of such an increased risk of developing cardiovascular events in patients with SARS-CoV-2 infection [8,9].

To this end, the cellular and biomolecular microvascular mechanisms of coagulopathy will yield crucial information on COVID-19-dependent thrombotic-derived systemic manifestations (i.e., thrombo-inflammation), thus paving the way for a more appropriate and targeted therapeutic strategy [10,11].

Recent exhaustive meta-analyses and critical literature reviews of hematologic, biochemical, and immunological biomarkers abnormalities associated with COVID-19 [12,13,14,15] revealed some paradigmatic patterns of laboratory biomarkers in patients with severe or fatal COVID-19, thus highlighting the role and function of well-known plasma biomarkers (e.g., cardiac troponins, C-reactive protein, cytokines and a plethora of metabolites), but also focusing attention on the pivotal role of excessive NET formation during COVID-19 progression, a process significantly contributing to the immuno-thrombotic state.

Over the past decades, several studies revealed a pathogenic role of NETs besides COVID-19 [4,16], encompassing various human diseases such as thrombo-inflammatory states, sepsis, trauma, lung, kidney, and nervous system injuries, cancer, and atherosclerosis, etc. [17,18,19,20]. The biomolecular characterization of NETs (NETome) identifies their main composition as an extracellular network of DNA, oxidant, and proteolytic enzymes of both cytosolic and granular origin, such as neutrophil elastase (NE), Myeloperoxidase (MPO), peptidyl arginine deiminase 4 (PAD4), cathepsin G, gelatinase, lysozyme C, leukocyte proteinase 3, lactoferrin, defensins, calprotectin, cathelicidins, HMGB1, actin and histones [3,21]. The release of these mediators, when not physiologically and finely regulated, has the potential to initiate and propagate inflammation and thrombosis, thus leading to both increased disease severity and shortened patient survival [18].

A plethora of recent studies underlining the roles and functions of extracellular histones as biomarkers for predicting outcomes of several human diseases have also been published [22,23]. Recent evidence especially demonstrates that the levels of NETs and histones may predict the cardiovascular risk [7], wherein circulating histones may function as signaling scaffold at the culprit site of myocardial infarction and stroke [24,25,26,27,28], also in COVID-19 patients with cardiac manifestations.

Interestingly, a recent study had first demonstrated that circulating histones play a crucial role in COVID-19-associated coagulopathy and mortality [29]. This outstanding and elegant research sheds light on the significant correlation between plasma histone levels and severity of COVID-19 infection, highly associated with severe coagulopathy, inflammation, and cardiac injury. In particular, the plasma levels of cardiac troponin were found to correlate with histone levels and were found to be significantly higher in COVID-19 patients who died compared to those who survived (median circulating histone levels in non-survivors vs. survivors: 29.6 μg/mL vs. 8.6 μg/mL, *p* = 0.002) [29], thus confirming literature data on both cytotoxic effects of extracellular histones on cardiomyocytes [25], and non-necrotic cardiac troponin release in COVID-19 patients [30].

Histones (i.e., positively charged multifunctional nuclear proteins) are key components in chromatin functions, which bind to the nucleosomal core particle around the DNA entry and exit sites. These intriguing molecules may be significantly released in body fluids during several targeted organ injuries (e.g., thrombosis, cancer, sepsis, etc.), thus mediating inflammatory pathways and coagulative cascade crucially linked to severity and mortality of many human pathologies [23,31].

All these observations suggest that injuries to the heart tissue caused directly by SARS-CoV-2 and/or indirectly by the release of histones SARS-CoV-2-related can be underlying causes of heart diseases (e.g., myocarditis and myocardial ischemia) in COVID-19 [25,29,32,33].

Our focused literature overview suggests that circulating extracellular histones may be significantly linked to cardiac injuries (at both cell- and tissue-level) [32], which are frequently reported in COVID-19 patients [33]. Thus, the well-known pro-inflammatory, pro-coagulant and cytotoxic functions of extracellular histones (released by NETs and nucleosome, acting as cytotoxic danger-associated molecular pattern, DAMP) [23,31] may represent an intriguing biomolecular mechanism that actively contributes to worsening the clinical course of COVID-19 and amplifying the risk of adverse outcomes [34].

In fact, histones exert endothelial and epithelial cytotoxicity interacting with both cell membrane phospholipids and cell membrane receptors (e.g., Toll-like Receptors, TLRs) and complement, thus promoting pro-inflammatory cytokine and chemokine release via MyD88, NFkB, and NLRP3 inflammasome-dependent pathway. Furthermore, histones could activate platelets, may bind red blood cells and increase their fragility, inducing phosphatidylserine exposure, finally promoting the development of micro-thrombi [29,31,34].

Moreover, they could be seen as a novel biomarker, which could assist risk stratification in patients with COVID-19 [29] and serve as a predictive factor for cardiac and lung injury/dysfunction, and ultimately are useful for individual management of the anticoagulant/anti-platelet therapy [35].

As concerns treatment options for COVID-19 coagulopathy [10], many studies have highlighted an urgent need to search pharmacological agents with endothelial-protective, histone-neutralizing properties and target histone removal in COVID-19 patients [36,37]. In particular, treatment of COVID-19 by heparins and heparinoids demonstrated their beneficial roles through complex biomolecular networks, based on both non-anticoagulant and anticoagulant mechanisms [38,39,40,41].

A recent medical hypothesis has also suggested that polycations (e.g., histones secreted by neutrophils following COVID-19) may worsen viral infections, that may be mitigated/counteracted by administration of negatively charged polyanionic drugs (e.g., heparins and heparinoids) [42].

According to this interesting hypothesis, in a frame of our studies, we observed a significantly different in vitro modulation of whole blood histone-induced inflammation and coagulation by several synthetic and natural heparins and heparinoids (glycosaminoglycan-based commercially available drugs) frequently used for COVID-19 treatment [10,37,39] (as recommended by WHO; https://www.who.int/publications/i/item/clinical-management-of-covid-19, accessed on 5 February 2022), such as unfractionated heparin, low molecular weight heparins (LMWH), danaparoid, fondaparinux, and sulodexide.

In addition to their well-known roles and functions as anticoagulants and pro-fibrinolytic compounds, their peculiar high negative charge density allows them to bind and strongly interact with several proteins and proteinases, revealing anti-inflammatory, anti-complement, immunomodulatory and anti-viral activities, independently to anticoagulant properties [38,39,40,41,43,44,45,46,47,48]. In particular, we have preliminarily found that the heparin and heparinoid formulations possess significantly different anti-inflammatory abilities and capabilities to bind/precipitate histones, and so ultimately to prevent histone-mediated cytotoxicity (unpublished observations).

Moreover, a recent report demonstrated that other polyanions (such as oligonucleotides mixture defibrotide) [45] might act as histone-neutralizing agents, thus blocking their pathological effects and protecting the endothelium from thrombo-inflammation.

Noteworthy, no studies are available on the role of heparins/heparinoids as histone-neutralizing agents in COVID-19 patients, despite the discovery of a novel role for histones in COVID-19 patients [29]. Interestingly, several clinical trials (currently more than one hundred studies registered in https://www.clinicaltrials.gov/ct2/results?cond=COVID-19&term=Heparin&cntry=&state=&city=&dist=, accessed on 21 March 2022) are describing the use and/or potential benefit of heparins in COVID-19, focusing attention on the anticoagulant effects of heparins in COVID-19 patients but neglecting the well-known non-anticoagulant biochemical property of heparin to prevent histone cytotoxicity [49].

All these observations suggest the urgent need to clinically evaluate the beneficial role of histone-neutralizing therapy by focused trials involving the interesting roles of polyanionic compounds as potential additional strategy for protecting tissues and organs from inflammatory, cytotoxic and procoagulant effects of circulating histones, implicated in myriad NET- and histone-accelerated disease states, and in COVID-19 complications (Figure 1).

Finally, in full agreement with the “multifactorial” definition of COVID-19 association with cardiac injury and multi-organ thrombo-inflammation [9], besides the imaging abnormalities and the systemic metabolic perturbations (e.g., hyper-inflammation and immuno-thrombosis), we will need to focus translational research and clinical trials to novel emerging laboratory biomarkers, such as circulating histones, which may induce multi-organ deleterious effects, explaining SARS-CoV-2 tropism and helping to refine cardiovascular and systemic risk stratification along with clinical management of COVID-19 patients [29,50] (Figure 1).

This landscape encourages the search for other pharmacological agents [38,48], also with endothelial-protective and histone-neutralizing properties in COVID-19 patients (e.g., apolipoprotein A–I, activated protein C, thrombomodulin, recombinant anti-histone IgG, peptidylarginine deiminases inhibitors, etc.) [51,52,53,54], but also to develop a circulating histone sensitive assay for laboratory medicine for better stratifying the risk of COVID-19 patients (as well as for sepsis-affected patients) [2,23].

All these promising future approaches reinforce the indefeasible urgent need for widespread (universal) vaccination against COVID-19, the primary strategy for lowering severity, morbidity, and mortality rate, and for limiting the diffusion and effectively protecting against COVID-19 variants.

## Figures and Tables

**Figure 1 jcm-11-01800-f001:**
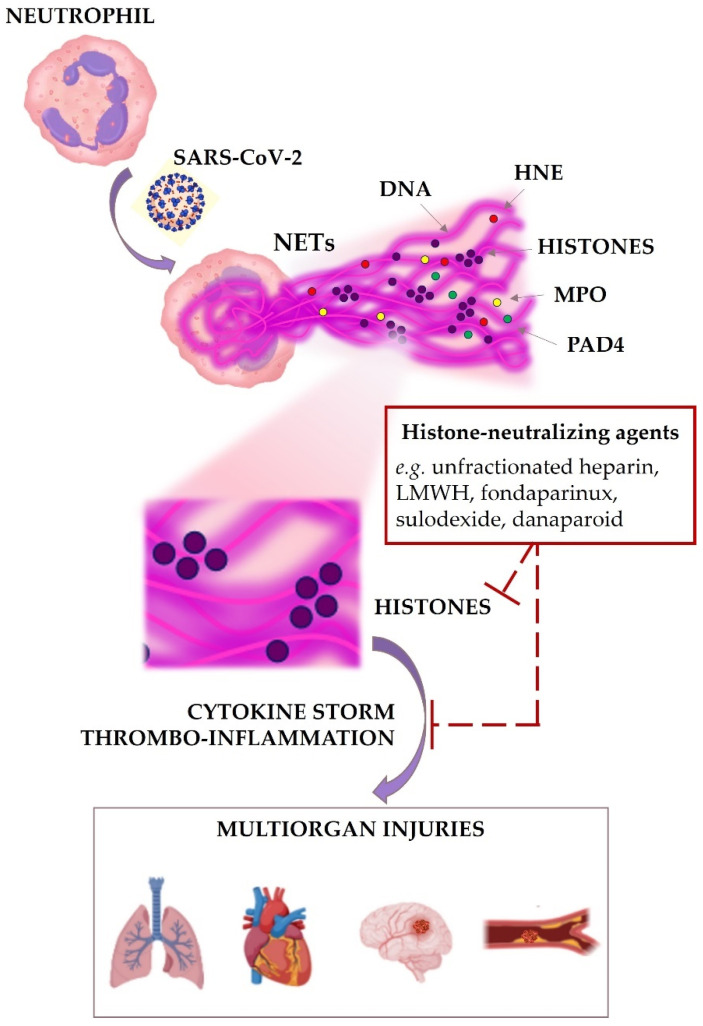
Histones as missing link among COVID-19 infection and multiorgan injuries. Among the main NET biomolecules, we focused attention on pathogenetic mechanisms of excess of histones potentially involved in multiorgan failure, coagulopathy, and systemic hyperinflammation during COVID-19 infection, and their possible therapeutic modulation by clinically used histone-neutralizing drugs (e.g., heparin and heparinoids). (NETs, neutrophil extracellular traps; DNA, deoxyribonucleic acid; HNE, human neutrophil elastase; MPO, myeloperoxidase; PAD-4, peptidylarginine deiminase-4; LMWH, low molecular weight heparins; HISTONES, positively charged multifunctional nuclear proteins).

## Data Availability

The datasets used in this paper are not publicly available since they are still under elaboration for publication by the Authors, but are avialable from the corresponding Authors on the reasonable request.

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
