# Peer review of "Do Circulating Histones Represent the Missing Link among COVID-19 Infection and Multiorgan Injuries, Microvascular Coagulopathy and Systemic Hyperinflammation?"

_jcm, 2022, doi:10.3390/jcm11071800_

Round 1
Reviewer 1 Report
The review “Do circulating histones represent the missing link among COVID-19 injection and multiorgan injuries, microvascular coagulopathy and systemic hyperinflammation published by Journal of Clinical Medicine aims to analyse the “in vitro” modulation of whole blood histone-induced inflammation and coagulation by several synthetic and natural heparins and heparinoids used for COVID treatment.
Using recent literature, the authors suggested new studies about the role of heparins/heparinoids as histone-neutralizing agents in COVID-19 patients, as the improvement of the knowlegement about no anticoagulant biochemical property of heparin to prevent histone cytostatic.
On the other hands, suggested new evaluate of the beneficial role of histone-neutralizing therapy, focused on polyanionic compounds, as a potential additional strategy to protect tissues and organs from inflammatory cytotoxic and procoagulant effects.
As a very important topic for COVID-19 treatment, the review is very interesting analyses, applied for clinical application.
Author Response
We would like to deeply thanks the Reviewer 1 for the comments and suggestions, highlighting her/his interest in this field.
Reviewer 1:
The review “Do circulating histones represent the missing link among COVID-19 injection and multiorgan injuries, microvascular coagulopathy and systemic hyperinflammation published by Journal of Clinical Medicine aims to analyse the “in vitro” modulation of whole blood histone-induced inflammation and coagulation by several synthetic and natural heparins and heparinoids used for COVID treatment.
Using recent literature, the authors suggested new studies about the role of heparins/heparinoids as histone-neutralizing agents in COVID-19 patients, as the improvement of the knowlegement about no anticoagulant biochemical property of heparin to prevent histone cytostatic.
On the other hands, suggested new evaluate of the beneficial role of histone-neutralizing therapy, focused on polyanionic compounds, as a potential additional strategy to protect tissues and organs from inflammatory cytotoxic and procoagulant effects.
As a very important topic for COVID-19 treatment, the review is very interesting analyses, applied for clinical application.
Author reply: We would like to thank the Reviewer for his/her comments and for the interest in this field.
Reviewer 2 Report
This is a very interesting article, focusing on a topic of key current importance in COVID-19 therapy. My comments are directed to help making the article have more impact on practitioners and researchers and more widely cited.
Neutrophil extracellular traps are commonly abbreviated as NETs, thus I recommend changing the abbreviation.
Authors should mention in the main text the principal proteins that are released in the scaffold of NETs.
Lines 66-68. “All these observations suggest that direct/indirect injuries to the heart tissue by both histones caused by SARS-CoV-2 can be underlying causes of heart diseases (e.g., myocarditis and myocardial ischemia) in COVID-19.”
Please explain this statement better.
There are some statements of correlations but no strength of the correlations is presented. Please provide data and strength of the correlations. You will thereby be providing data of great importance to readers. Some examples are:
- Line 52. “This outstanding and elegant research sheds light on the significant correlation between plasma histone levels and severity of COVID-19 infection, highly associated with severe coagulopathy, inflammation, and cardiac injury.”
- Line 55. “In particular, the plasma levels of cardiac troponin were found to correlate with histone levels and were found to be significantly higher in COVID-19 patients who died compared to those who survived…”
Line 71. “Thus, the well-known pro-inflammatory, pro-coagulant and cytotoxic functions of extracellular histones (released by NETs and nucleosome, acting as cytotoxic danger-associated molecular pattern, DAMP) …”
Please mention some mechanisms by which histones can induce pro-inflammatory, pro-coagulant and cytotoxic functions, some readers may not be familiar with it.
Line 109. Authors mentions that: “no studies are available on the role of heparins/heparinoids as histone-neutralizing agents in COVID-19 patients, despite the discovery of novel role for histones in COVID-19 patients…”
Since there are no studies, the authors may mention some of the clinical trials that are registered that use this drug, for example in clinicaltrials.gov there are about 30 clinical trials that study the use of some type of heparin.
Author Response
We would like to thank the expert Reviewer 2 for his/her comments and for the interest in this field. We fully agree and deeply appreciate the improvements suggested by the Reviewer 2.
Reviewer 2:
This is a very interesting article, focusing on a topic of key current importance in COVID-19 therapy. My comments are directed to help making the article have more impact on practitioners and researchers and more widely cited.
Author reply: We would like to thank the Reviewer for his/her insightful comments focused to improve the readability of the ms. Please, find below our replies.
- R) Neutrophil extracellular traps are commonly abbreviated as NETs, thus I recommend changing the abbreviation.
Author reply: Thanks for your suggestion. Accordingly, we modified the abbreviations throughout the text.
- R) Authors should mention in the main text the principal proteins that are released in the scaffold of NETs.
Author reply: We specified in the main the text the proteins, proteases, and molecules that have been mainly recognized in the scaffold of NETs.
- R) Lines 66-68. “All these observations suggest that direct/indirect injuries to the heart tissue by both histones caused by SARS-CoV-2 can be underlying causes of heart diseases (e.g., myocarditis and myocardial ischemia) in COVID-19.” Please explain this statement better.
Author reply: As reported in red in the main text, we modified the sentence as follow: “All these observations suggest that injuries to the heart tissue caused directly by SARS-CoV-2 and/or indirectly by the release of histones SARS-CoV-2-related can be underlying causes of heart diseases (e.g., myocarditis and myocardial ischemia) in COVID-19 [25,29,32,33]”.
There are some statements of correlations, but no strength of the correlations is presented. Please provide data and strength of the correlations. You will thereby be providing data of great importance to readers. Some examples are:
Line 52. “This outstanding and elegant research sheds light on the significant correlation between plasma histone levels and severity of COVID-19 infection, highly associated with severe coagulopathy, inflammation, and cardiac injury.”
Line 55. “In particular, the plasma levels of cardiac troponin were found to correlate with histone levels and were found to be significantly higher in COVID-19 patients who died compared to those who survived…”
Author reply: We modified these two sentences, sharing the same reference, as follow: “This outstanding and elegant research sheds light on the significant correlation between plasma histone levels and severity of COVID-19 infection, highly associated with severe coagulopathy, inflammation, and cardiac injury. In particular, the plasma levels of cardiac troponin were found to correlate with histone levels and were found to be significantly higher in COVID-19 patients who died compared to those who survived (median circulating histone levels in non-survivors vs. survivors: 29.6 μg/mL vs. 8.6 μg/ml, p=0.002) [29]….”
Line 71. “Thus, the well-known pro-inflammatory, pro-coagulant and cytotoxic functions of extracellular histones (released by NETs and nucleosome, acting as cytotoxic danger-associated molecular pattern, DAMP) …”
Please mention some mechanisms by which histones can induce pro-inflammatory, pro-coagulant and cytotoxic functions, some readers may not be familiar with it.
Author reply: We reported in red some mechanisms in the main text.
Line 109. Authors mentions that: “no studies are available on the role of heparins/heparinoids as histone-neutralizing agents in COVID-19 patients, despite the discovery of novel role for histones in COVID-19 patients…”
Since there are no studies, the authors may mention some of the clinical trials that are registered that use this drug, for example in clinicaltrials.gov there are about 30 clinical trials that study the use of some type of heparin.
Author reply: Thanks for your suggestion. As you recommended, we mentioned the presence of several clinical trials (currently more than one hundred studies registered in https://www.clinicaltrials.gov/ct2/results?cond=COVID-19&term=Heparin&cntry=&state=&city=&dist=) describing the use and/or potential benefit of heparins in COVID-19.